# Dichotomy of heavy and light pairs of holes in the $t$–$J$ model

A. Bohrdt[1,2], E. Demler[3] & F. Grusdt [4,5] ✉

A key step in unraveling the mysteries of materials exhibiting unconventional superconductivity is to understand the underlying pairing mechanism. While it is widely agreed upon that the pairing glue in many of these systems originates from antiferromagnetic spin correlations, a microscopic description of pairs of charge carriers remains lacking. Here we use state-of-the art numerical methods to probe the internal structure and dynamical properties of pairs of charge carriers in quantum antiferromagnets in four-legged cylinders. Exploiting the full momentum resolution in our simulations, we are able to distinguish two qualitatively different types of bound states: a highly mobile, meta-stable pair, which has a dispersion proportional to the hole hopping $t$, and a heavy pair, which can only move due to spin exchange processes and turns into a flat band in the Ising limit of the model. Understanding the pairing mechanism can on the one hand pave the way to boosting binding energies in related models, and on the other hand enable insights into the intricate competition of various phases of matter in strongly correlated electron systems.

Following the discovery of high-temperature superconductivity in the cuprates, understanding the mechanism by which pairs of charge carriers can form in a system with repulsive interactions has been a key question in the field, despite a general agreement that antiferromagnetic spin correlations play a prominent role[1–3]. Motivated by experimental results on the cuprate materials, a lot of theoretical and numerical work has focused on identifying the potential pairing symmetry[4,5] as well as the binding energies in these microscopic models[6,7]. Despite a vast research effort over several decades, the existence of a superconducting phase in the simplest model describing interacting electrons, the Fermi-Hubbard model, remains debated[8]. Competing orders, such as charge density waves and stripes, contribute to the difficulty in realizing as well as understanding superconductivity[9].

In order to unravel the competition between different orders, and thus the conditions for the existence of a superconducting phase, it is essential to gain a deeper understanding of the nature of individual pairs of charge carriers. The existence of pairs close to half-filling does not imply that for a finite density of holes, the system necessarily realizes a $d$-wave paired state. Instead, a finite number of charge carriers can for example self-organize into a charge or pair density wave state[10]. However, understanding whether and how pairs form in the two-hole problem is crucial to the subsequent understanding of the self-organization of many holes.

Here we approach the question of the underlying binding mechanism from an alternative perspective: through elaborate spectroscopic tools, we search for bound states of charge carriers in a quantum antiferromagnet and directly probe their internal structure. In particular, we numerically simulate rotational two-hole spectra, where different angular momenta can be imparted on the system, using time-dependent matrix product states. Crucially, these rotational spectra go beyond the standard pairing correlations through the momentum resolution they provide. The momentum dependence of the peaks in the spectral function enables direct insights into the

[1]ITAMP, Harvard-Smithsonian Center for Astrophysics, Cambridge, MA 02138, USA. [2]Department of Physics, Harvard University, Cambridge, MA 02138, USA. [3]Institut für Theoretische Physik, ETH Zurich, 8093 Zurich, Switzerland. [4]Department of Physics and Arnold Sommerfeld Center for Theoretical Physics (ASC), Ludwig-Maximilians-Universität München, Theresienstr. 37, München D-80333, Germany. [5]Munich Center for Quantum Science and Technology (MCQST), Schellingstr. 4, D-80799 München, Germany. ✉e-mail: fabian.grusdt@physik.uni-muenchen.de

effective mass of the pairs, which is an essential property for understanding their ability to condense at finite doping and temperature.

We study pairing between two individual holes doped into the two-dimensional $t$–$J$ model, which corresponds to the enigmatic Fermi-Hubbard model to second order in $t/U$ (up to next-nearest neighbor hopping terms, where $U$ is the on-site interaction) and describes electrons in cuprates[11]:

$$\hat{\mathcal{H}}_{t-J} = -t\,\hat{\mathcal{P}} \sum_{\langle i,j \rangle} \sum_{\sigma} \left( \hat{c}^{\dagger}_{i,\sigma} \hat{c}_{j,\sigma} + \text{h.c.} \right) \hat{\mathcal{P}} + \\ + J \sum_{\langle i,j \rangle} \hat{S}_i \cdot \hat{S}_j - \frac{J}{4} \sum_{\langle i,j \rangle} \hat{n}_i \hat{n}_j, \quad (1)$$

where $\hat{\mathcal{P}}$ projects to the subspace with maximum single occupancy per site; $\hat{S}_j$ and $\hat{n}_j$ denote the on-site spin and density operators, respectively. In our numerical simulations, we consider a 40 site long, four-legged cylinder, which is sufficiently long to ensure that the two-hole wavefront in the time evolution we consider below does not reach the edges of the system. This also means that the thermodynamic limit is essentially reached in the long direction, and our resulting spectra correspond to predictions at zero doping.

In order to probe a possible bound state of two charges, we consider an extension of conventional angle-resolved photoemission spectroscopy (ARPES). In particular, we excite the initially undoped antiferromagnet by creating not one, but two charges while simultaneously imparting angular momentum on the system. The resulting spectra thus directly contain information about the existence of possible bound states, their ground state energy, as well as their dispersion relation. In our numerical matrix product state calculations, we

find well-defined peaks in the rotational spectral function for all angular momenta, for spin-singlet as well as triplet pairs, and throughout an extended frequency range.

In order to gain a deeper understanding of the rotational two-hole spectra, we also consider the conceptually simpler $t$–$J_z$ model, where the $SU(2)$ invariant spin interactions are replaced by Ising-type interactions. Without additional spin dynamics, a direct comparison of our numerical results to an effective theory describing pairs of charge carriers bound by strings is possible, yielding excellent agreement in terms of the existence as well as the dispersion of the various bound states we observe. In particular, we discover a strongly dispersive low-energy peak, with a dispersion scaling with the hole hopping $t$, as well as completely flat bands at competitive energies. We attribute the flat bands to destructive interference of pairs with $d$-wave symmetry (See Supplementary Information file).

Upon introducing spin dynamics, the flat bands develop into weakly dispersive bands, whereas the $t$-dependent feature remains largely unchanged. We thus discover two qualitatively different kinds of bound states: highly dispersive peaks, including a high-energy feature with strong spectral weight in the $s$-wave spectra; and a weakly dispersive band, which has a high amount of spectral weight in the $d$-wave spectra. The dispersion of the latter is determined by the spin coupling $J$. The emergence of a slow time-scale set by $J$ is intuitive and well-known in the case of a single hole[12], which forms a spinon-chargon bound state and can thus only move as fast as the spin excitation[13]. In contrast, it is surprising to find a coherent bound state peak of two holes in the spectrum with a dispersion $\propto t$ extending over a wide range of energies without decaying into incoherent pairs of individual holes.

The remainder of this paper is organized as follows. We start by introducing the rotational two-hole spectra. We then discuss results for the $t$–$J_z$ model, where the $SU(2)$ invariant spin interactions are replaced by Ising-type interactions. We discuss the features found in the numerically obtained spectra in detail and compare them to a semi-analytical theoretical description of pairs of charge carriers[14]. Finally, we consider the full $t$–$J$ model.

## Results
### Rotational Spectra
In order to probe the internal structure of pairs of charge carriers, we study rotational spectra. We define an operator $\hat{\Delta}_{m_4}(j, \sigma, \sigma')$ that creates a pair of holes on the bonds adjacent to site $j$ with discrete $C_4$ angular momentum $m_4 = 0, 1, 2, 3$ as

$$\hat{\Delta}_{m_4}(j, \sigma, \sigma') = \sum_{i:\langle i,j \rangle} e^{im_4 \varphi_{i-j}} \hat{c}_{i,\sigma'} \hat{c}_{j,\sigma}, \quad (2)$$

with $\varphi_r = \arg(r)$ the polar angle of $r$; see Fig. 1a for an illustration. In order to annihilate a spin-singlet, we define the singlet pair operator (and similar for triplets) as

$$\hat{\Delta}^{(s)}_{m_4}(j) = \hat{\Delta}_{m_4}(j, \uparrow, \downarrow) - \hat{\Delta}_{m_4}(j, \downarrow, \uparrow). \quad (3)$$

The simplest term creating a spin-singlet excitation with discrete angular momentum $m_4$, charge two, and total momentum $k$ is directly given by the spatial Fourier transform of the singlet pair operator as

$$\hat{\Delta}^{(s)}_{m_4}(k) = \sum_j \frac{e^{-ik \cdot j}}{\sqrt{V}} \hat{\Delta}^{(s)}_{m_4}(j) \quad (4)$$

with volume $V$. The discrete angular momentum $m_4$ is a good quantum number at $C_4$ invariant momenta $k = (0, 0)$, $(\pi, \pi)$ only. Based on this

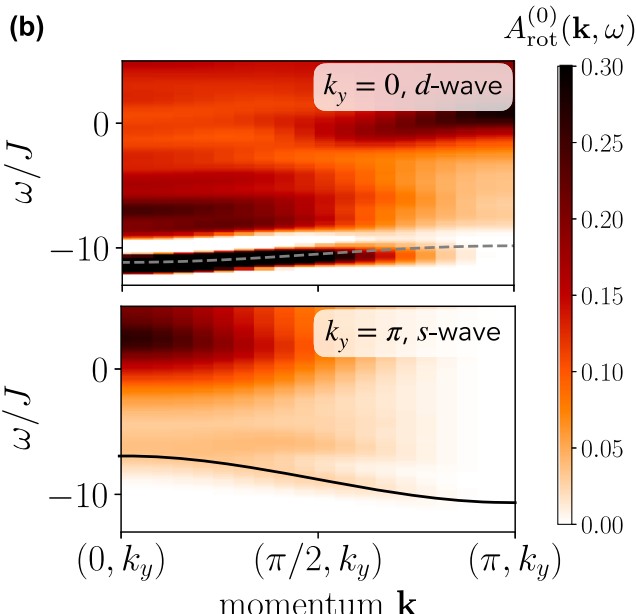

**(a)**

$$\left|\substack{\circ\circ\circ\\\circ\circ\circ\\\bullet\circ\bullet}\right\rangle + e^{im_4\pi/2}\left|\substack{\circ\circ\circ\\\bullet\circ\circ\\\circ\circ\bullet}\right\rangle + e^{im_4\pi}\left|\substack{\circ\circ\circ\\\circ\circ\circ\\\bullet\circ\bullet}\right\rangle + e^{im_4 3\pi/2}\left|\substack{\circ\circ\circ\\\circ\circ\bullet\\\bullet\circ\circ}\right\rangle$$

**(b)**

$A^{(0)}_{\text{rot}}(k, \omega)$

**Fig. 1 | Rotational spectroscopy of two holes in a singlet state in the $t$–$J$ model with $t/J = 3$, on a $40 \times 4$ cylinder, based on a time evolution up to $T_{\max}/J = 3$ and bond dimension $\chi = 1200$.** Energies are measured relative to the undoped parent antiferromagnet. **a** Sketch of the response probed by the rotational spectrum. **b** The upper (lower) plot corresponds to $k_y = 0(\pi)$ and a $d$-wave ($s$-wave) excitation. Data are shown as a function of momentum $k_x$ and frequency $\omega/J$. Gray dashed lines correspond to a cosine dispersion $-2J\alpha \cos(k_x) + b_J$, black line corresponds to a cosine dispersion $-2t\alpha \cos(k_x) + b_t$, where $\alpha = 0.33$ in both cases, $b_J = -11J$, and $b_t = -9J$.

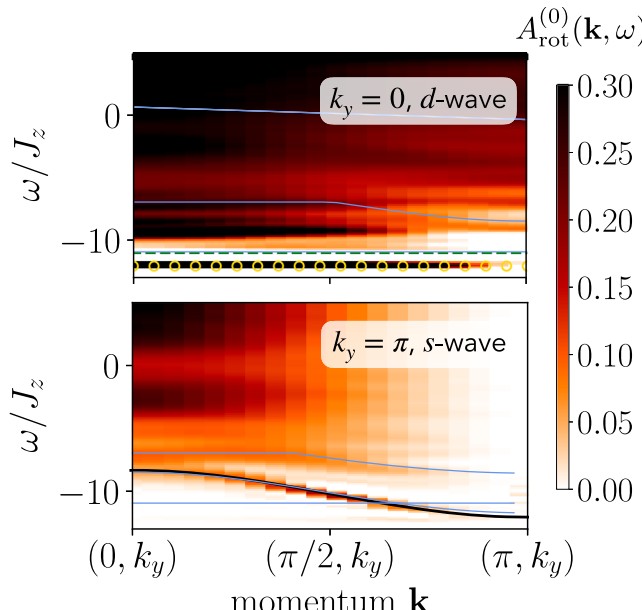

**Fig. 2 | Two-hole rotational spectra in the $t$−XXZ model for $t/J_z = 3$ and $J_\perp/J_z = 0.1$ on a 40 × 4 cylinder, based on time evolution up to $T_{max}/J_z = 10$ and bond dimension $\chi = 600$.** The colormap corresponds to numerical matrix product state simulations of the singlet two-hole rotational spectrum, blue lines are geometric string theory predictions for the position of states (all shifted by $-0.35J_z$), and the black line is a cosine fit. The upper (lower) plot corresponds to $k_y = 0$ ($k_y = \pi$) at $m_4 = 2$, $d$-wave ($m_4 = 0$, $s$-wave), and data are shown as a function of momentum $k_x$ and frequency $\omega/J_z$. In the top panel, the overall ground state energy for two holes is marked by orange circles for $k_y = 0$, and the green dashed line corresponds to twice the energy of a single hole (indicating a small pairing gap on the order of $J_z$).

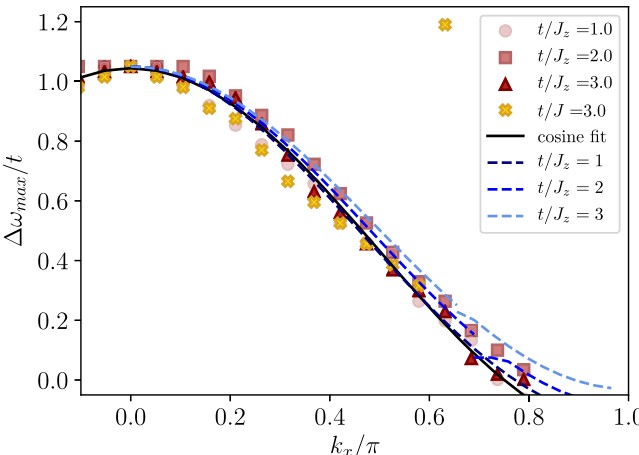

**Fig. 3 | Strongly dispersive pair state in the $t$−XXZ model for $t/J_z = 1, 2, 3$ and $J_\perp/J_z = 0.1, 1.0$ and $m_4 = 0$ ($s$-wave).** The symbols correspond to the position of the lowest energy peak at $k_y = \pi$ extracted from numerical matrix product state simulations of the singlet two-hole rotational spectrum. Yellow crosses correspond to the isotropic case, $J_z = J_\perp = J$ with $t/J = 3$. All data points are shifted vertically to collapse at $k_x = 0$. The blue dashed lines are geometric string theory predictions for the position of the strongly dispersing states. The black line is a cosine fit, $0.62\cos(k_x) + 0.72$, to the extracted peak positions for $t/J_z = 3, J_\perp/J_z = 0.1$.

operator, we now consider the rotational Green's function

$$\mathcal{G}_{rot}^{(m_4)}(\mathbf{k}, t) = \theta(t)\langle\Psi_0|\hat{\Delta}_{m_4}^{(s)\dagger}(\mathbf{k}, t)\hat{\Delta}_{m_4}^{(s)}(\mathbf{k}, 0)|\Psi_0\rangle, \tag{5}$$

which we calculate using time-dependent matrix product states[15–17]. The corresponding two-hole rotational spectrum, $-\pi^{-1}\text{Im}\mathcal{G}_{rot}^{(m_4)}(\mathbf{k}, \omega)$, in Lehmann representation is

$$A_{rot}^{(m_4)}(\mathbf{k}, \omega) = \sum_n \delta\left(\omega - E_n + E_0^0\right)\left|\langle\Psi_n|\hat{\Delta}_{m_4}^{(s)}(\mathbf{k})|\Psi_0^0\rangle\right|^2, \tag{6}$$

where $|\Psi_0^0\rangle$ ($E_0^0$) is the ground state (energy) of the undoped system and $|\Psi_n\rangle$ ($E_n$) are the eigenstates (eigenenergies) with two holes.

The two-hole rotational spectral function defined above is closely related to the dynamical pairing correlations frequently considered in the literature[8,18,19],

$$P(\omega) = \int dt\, e^{i\omega t}\left\langle\hat{\Delta}_{m_4}^{(s)\dagger}(t)\hat{\Delta}_{m_4}^{(s)}(0)\right\rangle, \tag{7}$$

where $\hat{\Delta}_{m_4}^{(s)} = \sum_j\hat{\Delta}_{m_4}^{(s)}(\mathbf{j})$. Here, however, we consider the full momentum dependence of the pairing correlations, which enables direct insights into the center-of-mass dispersion of pairs of charge carriers.

The resulting rotational spectra thus directly probe the existence of bound states and their internal structure: If a bound state of two holes with long-lived rotational excitations exists, the rotational spectra should exhibit well-defined coherent peaks. If on the other hand, such bound states do not exist, the excitation with the rotational operator $\hat{\Delta}_{m_4}(\mathbf{k})$ will lead to a broad continuum in the corresponding spectral function.

In Fig. 1b, we show the two-hole spectral function with angular momentum, i.e., $m_4 = 0$ ($s$-wave) and $m_4 = 2$ ($d$-wave) for the $t$−$J$ model

for momenta $0 \leq k_x \leq \pi$ and $k_y = \pi$ and $k_y = 0$, respectively. We find a well-defined coherent peak at low energies for all momenta, indicating the existence of a bound state. The spectrum furthermore reveals a plethora of different features, including a highly dispersive band (black line, $s$-wave excitation) as well as bands with a dispersion proportional to the spin-exchange $J$ (gray dashed lines, $d$-wave excitation). At momentum $\mathbf{k} = (\pi, \pi)$, the spectral weight vanishes for all energies for the $s$-wave excitation since $\hat{\Delta}_0^{(s)}(\mathbf{k} = (\pi, \pi)) = 0$.

In order to gain a deeper understanding of these intriguing results, we take a step back and analyze the conceptually simpler $t$−$J_z$ model in the following section.

## The $t$−XXZ model

We now consider a modification of the $t$−$J$ model, where the SU(2) invariant spin interactions are replaced by in-plane and Ising-type spin interactions with coupling constants $J_\perp$ and $J_z$:

$$\hat{\mathcal{H}}_{t-XXZ} = \sum_{\langle i,j\rangle}\left(J_\perp\left(\hat{S}_i^x\hat{S}_j^x + \hat{S}_i^y\hat{S}_j^y\right) + J_z\hat{S}_i^z\hat{S}_j^z\right) \\ - t\hat{\mathcal{P}}\sum_{\langle i,j\rangle}\sum_\sigma\left(\hat{c}_{i,\sigma}^\dagger\hat{c}_{j,\sigma} + \text{h.c.}\right)\hat{\mathcal{P}} - \frac{J_z}{4}\sum_{\langle i,j\rangle}\hat{n}_i\hat{n}_j. \tag{8}$$

In the limit of $J_\perp \ll J_z$, also called the $t$−$J_z$ model, the lack of spin dynamics facilitates our theoretical understanding. Experimentally, the anisotropic interactions can for example be realized by employing Rydberg interactions[20,21] or using ultracold molecules in tweezer arrays[21].

Remarkably, the two-hole spectral function, Fig. 2, exhibits a highly dispersive peak with a mass proportional to $1/t$, best identified at $k_y = \pi$ (bottom panel); I.e., we find a long-lived, tightly bound state of two holes, which can move as fast as the hole hopping $t$. This is in stark contrast to the case of a single hole in the same model, which has a very high effective mass $\gg 1/t$ and thus an almost flat dispersion[22], since it can only move due to Trugman loops[23], which are higher-order processes.

In Fig. 3, we further analyze the scaling of the mass of the pair by analyzing the position $\Delta\omega_{max}$ of the lowest energy peak at $k_y = \pi$ as a function of $k_x$ for different values of $t/J_z = 1, 2, 3$ and $J_\perp/J_z = 0.1$. Note that the frequency $\Delta\omega_{max}$ is defined relative to the energy of the highly

dispersive peak at momentum $\mathbf{k} = (0, \pi)$, and shown in units of the hole hopping $t$. We find a remarkable agreement in the overall shape of the dispersive peak for different values of $t/J_z$, thus highlighting the scaling with the hole hopping $t$.

The lowest-lying peak for $k_y = 0$ and $k_y = \pi/2$ is—within our numerical resolution—completely flat. Note that the situation of two unbound, and thus approximately immobile, holes should have a very small matrix element in the spectral function considered here, and therefore cannot account for the pronounced flat band peaks we find in the two-hole spectra. This is further corroborated by the direct comparison of the energy for two holes ($E_{2h} - E_{0h} = -12.08J_z$, orange circles in top panel Fig. 2) with twice the energy of a single hole ($2 \cdot (E_{1h} - E_{0h}) = -11.01J_z$, green dashed line in top panel Fig. 2): the latter is higher by $\approx J_z$, well above our spectral resolution.

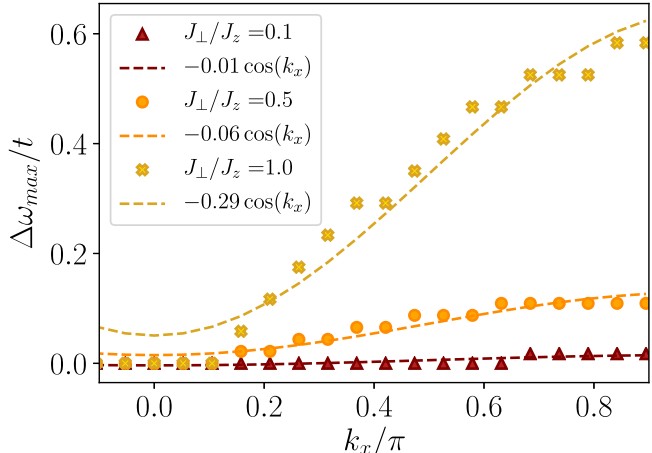

**Fig. 4 | Weakly dispersive pair in the $t$–XXZ model for $t/J_z = 3$, $J_\perp/J_z = 0.1$, 0.5, 1.0, and $m_4 = 2$ ($d$-wave).** The symbols correspond to the position of the lowest energy peak at $k_y = 0$ extracted from matrix product state simulations of the singlet two-hole rotational spectrum. All data points are shifted vertically to collapse at $k_x = 0$. Dashed lines are a cosine fit to the extracted peak positions with pre-factor as indicated in the legend, and additional offsets (not indicated).

In a companion paper[14], we extend the geometric string theory developed for a single hole[22, 24] to the case of two charge carriers. In particular, this geometric string theory approach describes the properties of two holes bound together by a string of displaced spins, and thus provides estimates of the energy and dispersion relation of such pair states, see blue lines in Figs. 2 and 3. Note that the existence of a state at a given energy does not imply that said state is visible in the spectral function, since the spectral weight, i.e. the overlap with the excitation we consider, can still be zero.

The geometric string theory correctly predicts the highly dispersive peak, as well as the existence of completely flat bands. Within this effective theory, the highly dispersive peak can be attributed to configurations where one hole re-traces the string created by the other hole, which allows the pair to move freely through the host anti-ferromagnet with an overall dispersion $\sim t$. The completely flat bands we find have also been predicted by an earlier theoretical study using a similar effective model[25]. We attribute them to the destructive interference of hole pairs with non-trivial rotational symmetry. A self-contained summary of the effective string theory is provided in the methods section.

## Results for the $t$–$J$ model

In the isotropic case, $J_\perp = J_z$, the strongly dispersive peak remains visible, see Fig. 1 bottom. In Fig. 3, we compare the peak position for the isotropic $t$–$J$ model (yellow crosses) with the $t$–XXZ model at $J_\perp/J_z = 0.1$ and find that the momentum dependence along the $x$-direction of the lowest-lying peak for $k_y = \pi$ is qualitatively very similar between the two cases. This indicates in particular that also in the $t$–$J$ model, a highly mobile, tightly bound pair state exists.

The flat bands, particularly visible in the $J_\perp/J_z = 0.1$ case at momenta $k_y = 0$ in the $d$-wave channel (and at $k_y = 0$, $\pi/2$ in the $d$- and for all $k_y$ in the $p$-wave channels, see Supplementary Figure 10), acquire a dispersion approximately proportional to $J_\perp^2/J_z$, as can be seen in the top plot in Fig. 1. We analyze this behavior in more detail by explicitly extracting the peak position $\Delta\omega_{max}$ at $k_y = 0$ for $t/J_z = 3$ and different values of $J_\perp/J_z$ in Fig. 4.

Again, the geometric string theory[14] correctly captures the highly dispersive peak with mass proportional to $1/t$ (black lines) discussed in Fig. 3. However, since this theoretical description does not account for

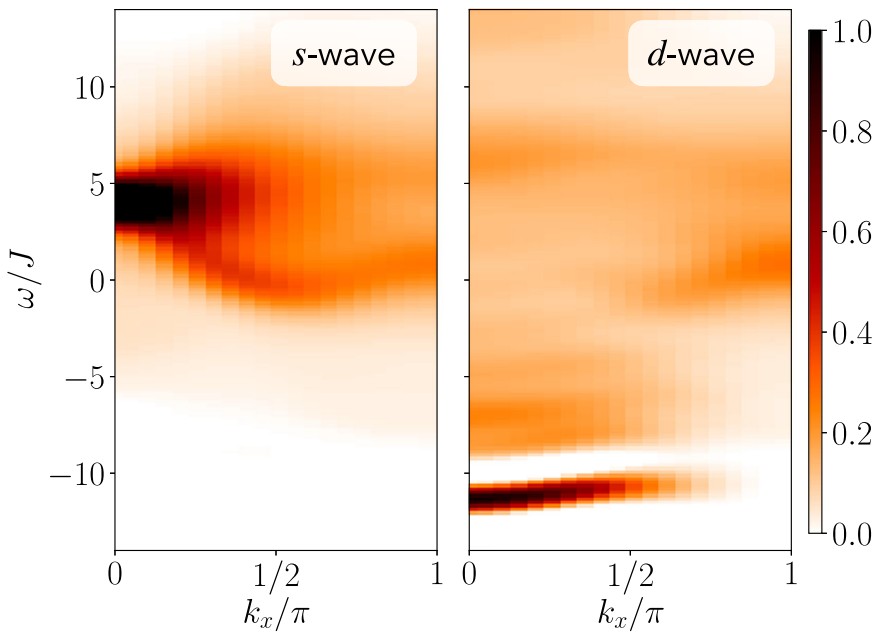

**Fig. 5 | Angular momentum dependence of the rotational two-hole spectra in the $t$–$J$ model.** We set $t/J = 3$ and $k_y = 0$, and for $m_4 = 0$, 2 (left, right column) calculated singlet two-hole spectra from time-dependent matrix product state simulations.

spin dynamics, it does not predict the dispersion proportional to $J_\perp^2/J_z$ (gray dashed lines). The corresponding bands in the $t$–$J_z$ model are flat, as predicted by the geometric string theory.

### Different angular momenta

In Fig. 5, we show the spectral function for the $t$–$J$ model at $t/J = 3$ and momentum $k_y = 0$ for $m_4 = 0, 2$, corresponding to $s$- and $d$-wave excitations. We find that the spectral weight exhibits a strong dependence on the angular momentum $m_4$ of the excitations. Before we have identified two qualitatively different types of bound states: a strongly dispersive band, see Fig. 3, and a weakly dispersive band, see Fig. 4. By considering different rotational excitations, we establish in Fig. 5 that the weakly dispersive pair, realizing the overall ground state within our spectral resolution, has almost exclusively spectral weight for the $d$-wave ($m_4 = 2$) excitation; for the $s$-wave ($m_4 = 0$) excitation, a large fraction of the spectral weight appears in a strongly dispersive high-energy feature. Both of these observations are also predicted by the geometric string theory[14].

Since the $40 \times 4$ cylinder geometry used in our numerical simulations weakly breaks the $C_4$ symmetry, $m_4 = 0$ and $m_4 = 2$ excitations can in principle hybridize. We find, however, that such hybridization is very weak with no significant mixing of spectral weight, see Fig. 5. This indicates weak finite-size effects in our calculations on 4-leg cylinders.

We note that in early exact diagonalization studies on small systems[18,19], the integration overall momenta, or the consideration of only momentum zero, leads to a much sharper low-energy peak in the $d$-wave than in the $s$-wave case. We attribute this to the widely different dispersions of the $s$- and $d$-wave peaks revealed here, and the exact $C_4$ symmetry in small systems.

The geometric string theory correctly predicts the $d$-wave character of the weakly dispersive band, as well as the accumulation of spectral weight at high energies for $\boldsymbol{k} = (0, 0)$[14]. A more detailed comparison of spectral weights for different values of $m_4$ shows very good qualitative agreement at low energies (see Supplementary Figures 1-3).

So far, we considered a singlet excitation. In order to annihilate a triplet instead, we define the triplet pair operator as

$$\hat{\Delta}_{m_4}^{(t)}(\boldsymbol{j}) = \hat{\Delta}_{m_4}(\boldsymbol{j}, \uparrow, \downarrow) + \hat{\Delta}_{m_4}(\boldsymbol{j}, \downarrow, \uparrow). \tag{9}$$

Upon considering the corresponding triplet spectral function, we find that the lowest-lying peaks are at higher energies than in the case of the singlet spectral function. This finding furthermore suggests that the lowest energy peak in the singlet spectral function cannot be attributed to the unbound states of two holes.

## Discussion

In this work, we extensively studied the properties of pairs of charge carriers in the $t$–$J$ and $t$–XXZ models through rotational spectra. We find well-defined coherent peaks at low energies for all momenta and angular momenta $m_4$. Our work provides an extensive numerical study of the mass of pairs of charge carriers in extended systems. We have revealed two qualitatively different types of bound states: First, a weakly dispersive peak with a dispersion approximately proportional to $J_\perp^2/J_z$, which has the most spectral weight for a $d$-wave excitation. Second, a highly dispersive peak, corresponds to tightly bound pairs with a mass proportional to $1/t$. We find the same signatures of these light pairs of charge carriers in the $t$–$J_z$ and the SU(2) invariant $t$–$J$ models. The bands corresponding to these light pairs, as well as the bands corresponding to heavy pairs, are qualitatively captured by a semi-analytic geometric string theory approach[14].

Our numerical studies of spectra are currently limited to four-leg systems since we cannot reach sufficiently long times to achieve the desired spectral resolution when working with larger bond dimensions required for six-leg systems. Finite-size effects can be expected to play a role, in particular on a quantitative level, but the good qualitative agreement of our results with predictions by the genuinely two-dimensional geometric string approach, along with the absence of significant hybridization of $d$ and $s$ wave excitations, support the view that four-leg cylinders are sufficient to capture key qualitative properties of hole pairs.

An intriguing direction for future research is the direct experimental probe of the two-hole rotational states. Understanding the pairing mechanism in the Fermi-Hubbard and related models has been one of the key motivations in the development of quantum simulators, and in particular cold atoms in optical lattices[26]. In the past two decades, remarkable progress has been made in the field[27,28], and several proposals to probe the pairing symmetry have been put forward[29,30]. More recently, the single-hole spectral function has been measured experimentally using ultra-cold atoms[31,32]. Using additional lattice modulations, the two-hole rotational spectral function considered here could be accessed experimentally. In solid state experiments, the $s$-wave two-hole spectral function can be accessed through coincidence angle-resolved photoemission spectroscopy[33], which relies on simultaneous measurements of two photo-electrons and provides direct insights into the pair Green's function. A different approach is based on Anderson-Goldman pair tunneling in a tunnel junction setup[34–36]: to study the structure of individual pairs in a strongly underdoped quasi-2D material as considered here, we propose to tunnel-couple the latter to a probe-superconductor along $z$-direction. Momentum resolution can in this case be obtained through an in-plane magnetic field.

The observation of light as well as heavy pairs in the spectra shown here furthermore suggests a real-space and -time experiment. Upon slowly releasing two holes next to each other in a cold atom experiment, a low-energy state of the pair can be prepared. In the ensuing time evolution, we predict the pairs to spread through the system in two distinct wave-fronts, corresponding to the light and the heavy pair, respectively. This phenomenology is expected more broadly, including in mixed-dimensional bilayer systems[37,38].

Finally, the existence of flat or weakly dispersive bands opens a new avenue to understand the many competing orders found experimentally in cuprate materials as well as numerically in Fermi-Hubbard and $t$–$J$ models at finite doping[8,39,40]. In the next step, we will investigate the dichotomy between two types of light and heavy pairs in the Fermi-Hubbard model. Furthermore, our work raises the interesting question of how the two types of tightly bound hole pairs discovered here relate to the Cooper pairs constituting high-temperature superconductors in copper oxides and whether they play any role in the pairing mechanism of the latter at all.

## Methods
### Geometric string theory

In order to interpret our numerically obtained two-hole spectra, we compare them to predictions by a simplified effective theory. The latter describes pairs of indistinguishable holes that are tightly bound by a geometric string of displaced spins. The detailed derivation and discussion of this two-hole geometric string theory can be found in ref. 14; see also ref. 25 for related earlier work. Here we will only provide an overview of the key structure, assumptions, and results of this theory. We emphasize that the main focus of the present article is on unbiased numerical results, which do not require major uncontrolled approximations beyond our choice of the microscopic model—in contrast to the effective geometric string theory.

The first key assumption of the geometric string theory is on the level of the employed Hilbert space. We consider exactly two holes, at positions $\boldsymbol{x}_{1,2}$ on the square lattice, and assume that for each state a unique string $\Sigma$ can be defined, composed of a sequence of string segments defined on the links of the lattice, which connects $\boldsymbol{x}_1$ to $\boldsymbol{x}_2$; i.e. $\boldsymbol{x}_2$ is uniquely defined by attaching $\Sigma$ to $\boldsymbol{x}_1$. By construction, we assume

that these string states $|\boldsymbol{x}_1, \Sigma\rangle$ span an orthonormal basis, i.e.

$$\langle \boldsymbol{x}_1', \Sigma' | \boldsymbol{x}_1, \Sigma \rangle = \delta_{\Sigma', \Sigma} \delta_{\boldsymbol{x}_1', \boldsymbol{x}_1}. \tag{10}$$

The basic motivation for working with this effective Hilbert space comes from considering two holes in a perfect Néel state in two dimensions: Starting from two neighboring holes, identified with the string-length $|\Sigma| = 1$ states, moving one hole away from the other creates a string-like memory of its trajectory, up to self-retracing paths, in the form of displaced Ising spins. For sufficiently short strings, the so-constructed truncated two-hole Hilbert space maps identically to the effective Hilbert space of the geometric string theory. For longer strings, this mapping no longer works due to loop effects, but their relative importance can be expected to remain small for sizable string lengths[22]. Going beyond the perfect Néel state, e.g., to the quantum Heisenberg antiferromagnet, even short-string states are not perfectly orthonormal due to quantum fluctuations, but it can still be expected that an orthonormalized basis with a similar structure can be constructed.

The second key assumption is on the level of the effective Hamiltonian. In the geometric string theory, we include hole tunneling by including processes where the string is extended or retracted by one segment on either end. Moreover, we take into account spin-spin couplings indirectly, through a string potential $V_\Sigma$; we further assume that the latter depends only on the length $\ell_\Sigma$ of the string $V_\Sigma \propto \ell_\Sigma$, with a pre-factor that can be determined from the case of straight strings[22]. This string potential models the energy cost associated with the frustrated bonds created by the motion of the holes through the host antiferromagnet.

Finally, to solve the effective string model, we further truncate the basis by taking into account only the lowest-lying rotational excitations[22] of the strings and restricting their overall length. Making use of the conservation of the pair's center-of-mass momentum, we can derive the momentum-resolved low-energy excitation spectrum of the tightly bound pairs[14]. This leads to the line shapes shown in Figs. 2 and 3 of the main text.

Since the geometric string theory, per construction, correctly captures the short-length strings, it provides a natural phenomenological theory to employ for the description of two-hole spectra: The latter characterize tightly bound two-hole eigenstates featuring a sizable overlap with string-length one states at a given total momentum $\boldsymbol{k}$ and $C_4$ angular momentum $m_4$. Other, much more loosely bound, paired states of individual magnetic polarons, could also lead to features in the two-hole spectrum, although with reduced spectral weight due to their overall size. Such states, however, can neither be described by the geometric string theory nor do we find any clear signatures for them in our numerically obtained spectral functions.

The central predictions of the effective geometric string theory entail[14]: (i) The existence of two types of tightly bound hole states, namely a highly dispersive set of states with bandwidth $\propto t$; and a completely flat set of states with zero bandwidth originating from destructive interference effects, see supplements. (ii) A distribution of spectral weights in the two-hole spectra which is in good qualitative agreement with our numerical observations; this includes in particular the prediction of flat $d$-wave pairs at low energies and a pronounced high-energy feature in the spectrum at $\boldsymbol{k} = 0$ in the $s$-wave channel—see supplement for comparisons of our numerics with the effective theory. (iii) The association of $d$-wave pairing symmetry with flat, or weakly dispersing, bands; and the association of $s$-wave pairing symmetry with highly dispersive bands corresponding to light hole pairs.

These features can also be observed in our unbiased numerical calculations of the two-hole spectra. The effective geometric string theory thus allows us to interpret our numerical findings in an intuitive way, supporting our claim that long-lived tightly bound paired states of holes exist in the two-dimensional $t$–$J$ model. The underlying pairing is facilitated by the frustrating effect of strings, formed directly through the hole motion in the host antiferromagnet.

## Data availability
All presented data are available from the authors upon request.

## Code availability
The used numerical code (TenPy Package) is publicly available at refs. 41,42.

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

## Acknowledgements
We thank Immanuel Bloch, Antoine Georges, Markus Greiner, Mohammad Hafezi, Lukas Homeier, and Ulrich Schollwöck for fruitful discussions. This research was funded by the Deutsche Forschungsgemeinschaft (DFG, German Research Foundation) under Germany's Excellence Strategy—EXC-2111—390814868 (A.B., F.G.), by the European Research Council (ERC) under the European Union's Horizon 2020 research and innovation programm (Grant Agreement no 948141)—ERC Starting Grant SimUcQuam (F.G.), by the ARO—grant number W911NF-20-1-0163 (E.D.), and by the NSF through a grant for the Institute for Theoretical Atomic, Molecular, and Optical Physics at Harvard University and the Smithsonian Astrophysical Observatory (A.B.).

## Author contributions
A.B. performed all numerical simulations. F.G. performed calculations for the geometric string theory. A.B., E.D., and F.G. contributed to the analysis of the results and the writing of the manuscript.

## Funding

## Competing interests
The authors declare no competing interests.
