## [Peer Review File · Nature Communications]

REVIEWER COMMENTS

Reviewer #1 (Remarks to the Author):

The manuscript by Bohrdt, Demler and Grusdt addresses the properties of pairs of charge carriers in the classic $t - J$ model (and its $t - XXZ$ extension) using sophisticated numerical time-dependent matrix-product computations, focusing specifically on the "rotational spectra". They report numerical evidence for two qualitatively different types of bound states involving light and heavy pairs. Given the enormous progress that the experimental field of quantum gas microscopy has had in recent years and the impact that even previous work from these authors have had on the field, the present manuscript is a timely addition to a large pre-existing body of work. I think the manuscript can be accepted for publication in Nature communications, but it would benefit the readers if the authors addressed the following questions and comments:

- 1) To make this manuscript somewhat self-contained (and if the length constraints allow for it), perhaps the authors want to include a short subsection (summary from the cross-cited manuscript?) on the comparisons of their numerical data to "geometric string theory". I am certain a fair number of interested readers who are not actively working on the subject will not be aware of this formalism, and a brief exposition might be useful.
- 2) While the discussion surrounding the pure Ising limit is useful and has its advantages, I wonder if the pure XY limit is interesting as well? Or is it almost identical to the Heisenberg limit? I imagine the finite temperature properties associated with the Heisenberg vs. XY limits are quite different.
- 3) While the numerical computations for the 4-legged ladders cylinders are certainly impressive, it would be useful to have some explicit remarks/results on the robustness of the conclusions with varying numbers of legs.
- 4) Given the ambitious remarks in the abstract and introduction on how studying the present problem might help boost the pairing scale and the problem at finite doping, it might be useful to comment (even speculate?) on what these computations have really taught us in terms of the big picture. Similarly, is it possible to use the present results (or perhaps consider extensions of this setup) to comment on the origin of the near-degeneracy of competing orders in the pairing and density-wave channels observed both in numerics and in the cuprates?

Reviewer #2 (Remarks to the Author):

The authors study pairing of two holes doped into the four-legged t-J model and report two qualitatively different types of bound states: a highly mobile pair with a dispersion scaled by the hopping t and a heavy pair with that scaled by J_{\perp}^2 / J_z .

While I do not find the present work significant and interesting as a potential paper in Nature Communications, this negative assessment might be due to many concerns and questions. Therefore I would first suggest that the authors address all of them in a convincing way in the manuscript.

- Two holes can in principle individually move with a hopping amplitude t , but the pairing state has a dispersion given by J . This is nothing but the physics in the t-J model, which is already clear without calculations.

- Both t and J are largely renormalized by a factor of $\alpha=0.33$. From a view of the physics of the t-J model (close to zero doping), I would expect that the renormalization of t and J should be different, that is, it is only t that is largely renormalized whereas the renormalization of J is much weaker than t . What could be the physical reason to have the same factor of α in front of t and J ?

- Is the hole doping rate $p=2/(4 \times 40) = 1.25\%$? Does the system retain antiferromagnetic order? Do the authors consider a coexistence state of pairing and antiferromagnetism? Answers should be given in the main text.

- How does one understand the absolute energy in, for example, Fig.1? I first assumed that this energy should be related with the pairing gap energy, but this should not be the case since the energy scale is too high in Fig.1. A physical understanding of the absolute energy should have been given.

- Rather the width of the dispersion scaled by J seems to be related with the pairing gap amplitude. On the other hand, the width of the dispersion scaled by t may not have a physical meaning of the pairing gap. A physical understanding of the dispersion width should have been given.

- In the lightly doped t-J model, the Fermi surface may appear as a hole pocket around $(\pi/2, \pi/2)$. Therefore one would expect that pairing should occur there, which is, however, not the case in Fig.1. Why?

- Just below Fig.2 the authors state "Remarkably, the two-hole spectral function, Fig.2, exhibits a highly dispersive peak with a mass proportional to $1/t$." I am not sure which panel in Fig.2 the authors have in mind. $k_y=\pi$?

- What is ω_{\max} ? It can be positive in Fig.3 and negative in Fig.4.

- I do not figure out a reason why the lowest lying peak for $k_y=0, \pi/2$ becomes completely flat whereas that for $k_y=\pi$ does not.

- In Fig.3 I do not see "yellow crosses".

Note to both Referees:

While working on the referee replies, we became aware of a small bug in the evaluation of the rotational spectra (the raw data was correct and has not been changed, but to calculate the rotational Green's function different contributions need to be added up with correct phases, and this is where a small numerical mistake was previously made which has now been fixed).

As a result, the comparison of our unbiased numerics based on time-dependent matrix product states with the effective geometric string theory has significantly improved, as can be seen in the updated plots of the revised manuscript.

Our conclusions in the main text are not affected: The bug erroneously suggested more hybridization between different rotational excitation branches than there really are. As such, the positions of spectral lines did not change, but the distribution of spectral weights did. Our key figures 3 and 4, where we establish the different types of dispersions present in the system, are completely unchanged. Moreover, our conclusion that the weakly dispersive band is associated with an internal d-wave symmetry of the pair has been further corroborated by the corrected data: The comparison of d- and s-wave spectra in Fig. 5 shows a much more striking difference at low energies now. We provide the old (erroneous) new (correct) figures at the bottom of this page for your convenience and comparison.

We apologize for the delay in responding to your reports, and appreciate your inputs on our manuscript. The delay on our end was caused by the time it took to fix the above-mentioned bug and double-check that no further errors are present, which we have carefully confirmed.

Below we provide detailed answers to all comments and criticism by the referees. We have addressed all points, and are convinced that our work will attract significant attention in the future. We hope you share our enthusiasm about the (now even clearer) results, and that you will recommend publication of our work in Nature Communications.

Sincerely, the authors,

Annabelle Bohrdt
Eugene Demler
Fabian Grusdt

Old Fig. 5 (erroneous)

Revised Fig. 5 (correct)

Reviewer #1 (Remarks to the Author):

The manuscript by Bohrdt, Demler and Grusdt addresses the properties of pairs of charge carriers in the classic $t - J$ model (and its $t - XXZ$ extension) using sophisticated numerical time-dependent matrix-product computations, focusing specifically on the "rotational spectra". They report numerical evidence for two qualitatively different types of bound states involving light and heavy pairs. Given the enormous progress that the experimental field of quantum gas microscopy has had in recent years and the impact that even previous work from these authors have had on the field, the present manuscript is a timely addition to a large pre-existing body of work. I think the manuscript can be accepted for publication in Nature communications, but it would benefit the readers if the authors addressed the following questions and comments:

Our reply:

We thank the referee for carefully evaluating our manuscript, putting our work in context, and in particular for recommending publication in Nature communications. We have addressed the additional points raised by the referee, which have strengthened our manuscript and made it more accessible to the broad readership of Nature Communications.

Referee:

1) To make this manuscript somewhat self-contained (and if the length constraints allow for it), perhaps the authors want to include a short subsection (summary from the cross-cited manuscript?) on the comparisons of their numerical data to "geometric string theory". I am certain a fair number of interested readers who are not actively working on the subject will not be aware of this formalism, and a brief exposition might be useful.

Our reply:

We thank the referee for this excellent suggestion. Following their advise, we have included a section in the Methods part where we provide a self-contained summary of the main assumptions and results of the two-hole geometric string theory.

Referee:

2) While the discussion surrounding the pure Ising limit is useful and has its advantages, I wonder if the pure XY limit is interesting as well? Or is it almost identical to the Heisenberg limit? I imagine the finite temperature properties associated with the Heisenberg vs. XY limits are quite different.

Our reply:

We thank the referee for this interesting question. In general we expect that the physics in the t -XY model, with vanishing Ising couplings, should be qualitatively similar to what we observe in the t -J model, as long as one considers the sector with $S_z_{\text{tot}}=0$. Namely, the undoped parent antiferromagnet (AFM) has an ordered ground state at zero temperature in $d=2$ dimensions and strong AFM correlations on nearest-neighbor bonds. The latter give rise to a non-zero string tension, similar to the t -J and t -XXZ models, albeit with quantitative differences. Since we haven't performed any quantitative simulations of this limit so far, we prefer not to discuss this case in our manuscript.

The referee also raises the interesting point of potential differences at finite temperature, $T>0$. Indeed, the XY model can support quasi-long range power-law correlations below the non-zero BKT transition temperature $T<T_{\text{BKT}}$, in contrast to the $SU(2)$ invariant model where any non-zero temperature gives rise to exponentially decaying AFM correlations. However, for the formation of

bound holes connected by a confining string, only near-neighbor (i.e. local) correlations play a role: When the holes move through the surrounding AFM, they locally distort the AFM anti-alignment, which causes an energy cost of the geometric string proportional to its length. Therefore whether AFM correlations at long distances decay exponentially or as a power-law should not change the essential structure of the pairs that we predict. Quantitative numerical studies of this regime are beyond our current computational capabilities, however, and hence we prefer not to make any claims in our manuscript.

Referee:

3) While the numerical computations for the 4-legged ladders cylinders are certainly impressive, it would be useful to have some explicit remarks/results on the robustness of the conclusions with varying numbers of legs.

Our reply:

We thank the referee for bringing up this important point and appreciating our numerical efforts. Of course we agree that larger six-leg systems would be desired to reassure us of the qualitative correctness of our findings on the currently achievable four-leg cylinders. Quantitatively, finite-size effects can certainly be expected to play a role.

Ultimately, only significantly extended numerical studies can settle this issue, but a few points make us optimistic: (i) In single-hole studies of spectra on four-leg cylinders, finite-size effects were confirmed to be small and of purely quantitative nature, by comparing with large-scale Monte-Carlo studies in more extended two-dimensional systems, see [Bohrdt et al., PRB 102, 035139 (2020)] for a detailed discussion; (ii) our qualitative results are robust when tuning from a strong Ising anisotropy to SU(2) invariant Heisenberg interactions. If finite-size effects are large, or even dominant, one may expect them to depend on the details of the underlying spin-spin couplings, which does not appear to be the case; (iii) qualitatively our results agree with predictions by the geometric string theory, which is a genuinely two-dimensional approach. Since the geometric string theory provides a rather phenomenological description of the system this cannot be viewed as a proof of any sort, but at least it provides a positive consistency check of our results against finite-size effects; (iv) the use of a narrow 4x40 cylinder in our simulations breaks the C4 symmetry, which can in principle lead to a hybridization of states with m_4 and $m_4 \pm 2$ due to finite-size effects. With the bug fixed in our numerics (see first page of this document), we observe only extremely weak hybridization of s- and d-wave pairs, which further indicates that finite-size effects are weak.

We agree with the referee that this is an important point, relevant to our readership. Hence we decided to add a paragraph to the summary & outlook section discussing this issue: "*Our numerical studies of spectra are currently limited to four-leg systems, since we cannot reach sufficiently long times to achieve the desired spectral resolution when working with larger bond-dimensions required for six-leg systems. Finite-size effects can be expected to play a role, in particular on a quantitative level, but the good qualitative agreement of our results with predictions by the genuinely two-dimensional geometric string approach, along with the absence of significant hybridization of d and s wave excitations, support the view that four-leg cylinders are sufficient to capture key qualitative properties of hole pairs.*"

Referee:

4) Given the ambitious remarks in the abstract and introduction on how studying the present problem might help boost the pairing scale and the problem at finite doping, it might be useful to

comment (even speculate?) on what these computations have really taught us in terms of the big picture. Similarly, is it possible to use the present results (or perhaps consider extensions of this setup) to comment on the origin of the near-degeneracy of competing orders in the pairing and density-wave channels observed both in numerics and in the cuprates?

Our reply:

The referee raises a fair and important question. We prefer not to speculate too much about this topic in our manuscript — keeping in mind how contentious and emotionally the entire topic surrounding the pairing mechanism in high- T_c superconductors has been discussed for several centuries already. That said, we are more than happy to provide our own perspective in our reply to the referee. If the referee and editors feel strongly that such a discussion should be included in our manuscript before publication, we could agree to add it.

Broadly speaking, our results support the view that magnetic polarons, i.e. the individual hole-type charge carriers, pairs of holes, and excitations such as Hubbard-Mott excitons constituted by bound doublon-hole pairs are closely related objects featuring a rich internal structure. The meson-like character of these charge carriers / excitations qualitatively explains their emergent internal structure, such as their rich and distinct dispersion relations which we revealed here. Our findings point towards emergent universal aspects shared by these constituents, and suggest new experimental probes. For example, resolving the internal angular momentum of the emergent mesons allows to reveal the different internal states; using the rich ultracold atom toolbox, direct observations of the internal string-like structure in real-space are also possible.

More specifically, our key findings are as follows:

The first key lesson we have learned from our calculations is the existence of long-lived, meta-stable hole-pairs, as revealed by our prediction of several sharp features in the two-hole spectral function, including up to high energies $O(t)$. This is a very significant finding, strongly supporting the point of view that doped holes in an antiferromagnet feature rich internal structure. Remarkably, no such calculations in large-scale systems with full momentum-resolution have been performed before.

Second, we revealed two qualitatively distinct types of pairs: Heavy pairs with a relatively flat dispersion, and highly mobile pairs with a band-width on the order of the tunneling t . The latter is particularly remarkable, given that individual holes are well-known (since many decades) to be heavy, featuring a bandwidth on the order of the super-exchange J . Likewise, in single-electron/ hole spectral studies (i.e. ARPES) of under-doped cuprates, no coherent features on the order of t are observed. Hence our study reveals a new type of constituent built from two tightly bound holes (as we know from its sizable spectral weight), which we predict to exist in the general parameter-regime relevant to high- T_c cuprate superconductors.

Third, the co-existence in the spectrum of heavy and light bound states of holes — the dichotomy we revealed here — points to some interesting microscopic physical effects might possibly be involved in understanding competing orders observed in cuprates. Indeed, we find that heavy pairs of holes with d-wave character are similarly low in energy as light pairs of holes with s-wave character. The closeness in energy of such pairs was known before, but not their vastly different masses. This is relevant, since one naturally expects heavy constituents to be more amenable to localization due to interaction effects, which leads to the question whether the observed behavior may be related to pair-density wave structures observed in some strongly correlated materials. On the other hand, the heavy d-wave pairs would naturally be expected to relate to the d-wave superconductivity observed in cuprates.

This brings us to the final point: In a follow-up work we are currently working on a strong-coupling perspective on high-temperature superconductivity. The basic idea starts from the experimental observation that superconductivity is BCS-like for all observed dopings in the cuprate superconductors, with clear signatures of (large and small) Fermi-surfaces, see e.g. [Sous & Kivelson, arXiv:2210.13478]. We then note that the relevant unpaired doped holes can interact by virtually recombining into the tightly bound pairs of holes whose structure we studied in the present paper — much like around Feshbach resonances widely used in atomic physics to tune interactions. Since the heavy paired d-wave state is slightly higher in energy, it can *mediate strong attractive d-wave interactions* between the individual doped holes, explaining the robust emergence of d-wave pairing at strong coupling / low doping. The s-wave channel, in contrast, is associated with highly mobile holes — due to their large band-width of order t that we revealed in the present manuscript. As we show in our follow-up work, this explains why s-wave attraction cannot be mediated in a similar fashion. Such considerations rely heavily on understanding the structure of tightly bound pairs, such as their center-of-mass dispersion, which we revealed here for the first time.

How far this new perspective — kicked off by our present manuscript — will get us remains to be seen, but our work certainly makes new predictions about the structure of paired holes that can be experimentally confirmed or refuted and thereby teach us about the microscopic origins of pairing in the Hubbard model and its descents.

Reviewer #2 (Remarks to the Author):

The authors study pairing of two holes doped into the four-legged t-J model and report two qualitatively different types of bound states: a highly mobile pair with a dispersion scaled by the hopping t and a heavy pair with that scaled by J_{\perp}^2/J_z .

While I do not find the present work significant and interesting as a potential paper in Nature Communications, this negative assessment might be due to many concerns and questions. Therefore I would first suggest that the authors address all of them in a convincing way in the manuscript.

Our reply:

We thank the referee for their time, and for clearly stating all their concerns and questions which we address below. We hope we can convince the referee of the significance of our findings, such that they can join referee #1 in recommending our work for publication in Nature Communications.

Referee:

- Two holes can in principle individually move with a hopping amplitude t , but the pairing state has a dispersion given by J . This is nothing but the physics in the t-J model, which is already clear without calculations.

Our reply:

We would like to clarify that **one of the pairing states we reveal has a robust large dispersion corresponding to a hopping amplitude t** , both in t-J and t-Jz models. I.e. we disagree with the statement by the referee that "*the pairing state has a dispersion given by J* ". As emphasized throughout our paper, we reveal two distinct types of paired states, with dispersions scaling as either directly $\propto t$ or as $\propto J_{\perp}^2/J_z$.

As discussed in the manuscript, and further below, both features can be directly understood from the parton / meson picture: In the latter, the single hole is viewed as a bound/ confined state of a heavy (mass $\sim 1/J$) spinon and a light (mass $\sim 1/t$) chargin; hence its dispersion scales as $\sim J$, as pointed out by the referee. Likewise, however, the two-hole state can be viewed as a bound / confined state of two light charginos (but *no* spinons), which can hence feature a dispersion $\sim t$.

Let us put this in context and emphasize the significance of our finding:

- (i) There is a microscopic tunneling term $\propto t$ in the t-J model — we believe this is what the referee refers to when they state that holes can "*in principle individually move with a hopping amplitude t* ". However,
- (ii) the mere existence of this large tunneling scale does *not* mean that it directly gives rise to a coherent feature in the spectrum (i.e. a corresponding long-lived, at least meta-stable, excitation). Indeed, this is the well-known phenomenology of the t-J model beyond $d=1$ dimension: the single-particle spectrum shows only a coherent quasiparticle peak featuring a dispersion $\propto J$ — we believe this is what the referee refers to when they state that this is "*already clear without calculations*". No coherent features on a scale $\propto t$ are visible in the single-particle spectrum. Importantly,
- (iii) we demonstrate the **existence of a coherent quasiparticle peak** in the two-hole spectral function, corresponding to a long-lived meta-stable pair excitation, **which does indeed**

feature a dispersion $\propto t$. The existence of such a peak is remarkable, and has not been directly revealed before in any large-scale numerical study, because naively one would expect that such a paired state should immediately decay into pairs of individual holes, each featuring a weaker dispersion $\propto J$. This does not appear to be the case in our numerics. Finally, (iv) we reveal a second type of paired state in the two-hole spectrum, which features a narrow dispersion $\propto J_1^2/J_z$ becoming entirely flat in the Ising limit. We agree with the referee that the emergence of such narrow bands per-se is not surprising in the 2D t-J-type models. However, the shape of the pair-dispersion itself has not been previously known (it cannot simply be derived from the well-known one-hole dispersion), and in our work we compare to semi-analytical predictions by effective theories, which provides important physical insights and understanding of the resulting paired states. These details are essential to develop a comprehensive theory of hole-pairing, and cannot be obtained without detailed calculations like the ones we performed here.

Referee:

- Both t and J are largely renormalized by a factor of $\alpha=0.33$. From a view of the physics of the t-J model (close to zero doping), I would expect that the renormalization of t and J should be different, that is, it is only t that is largely renormalized whereas the renormalization of J is much weaker than t . What could be the physical reason to have the same factor of α in front of t and J ?

Our reply:

We thank the referee for raising this excellent question.

When they state their expectation would be that t should be largely renormalized while J should only be weakly renormalized close to zero doping, we believe the reviewer again refers to the commonly known phenomenology of the 2D t-J model that the dispersion of a single dopant reduces significantly to $\propto J$ instead of $\propto t$ as one might naively expect.

In applying this analogy to the two-hole case we study here, it is assumed that first each hole forms a new low-energy quasiparticle with a strongly renormalized dispersion, down to $\propto J$, and only then pairs can form. In such a picture it would indeed be remarkable to find any features scaling $\propto t$, and any renormalization of t and J should be independent from each other.

The physical picture we have is a drastically different one: We believe that paired states of holes connected by a string can form which can move completely freely and coherently through the antiferromagnet (AFM) with a dispersion $\propto t$. The basic mechanism, namely one hole repairing the string created by the other, has long been discussed in the community, but it has never been firmly established that this can indeed give rise to long-lived coherent pair excitations with a large bandwidth as we reveal here. Notably, this mechanism completely invalidates the more commonly used picture discussed above: We argue that one should **not** think of $\propto t$ couplings as being quickly renormalized down to scales $\propto J$, but rather that meta-stable long-lived structures with dispersion $\propto t$ can directly lead to the formation of eigenstates spanning a large range of energies, in excellent agreement with the numerical findings in our manuscript. This is the essence of the parton / meson picture we mentioned previously.

This finally brings us back to the original question posed by the referee: Why are the features we observe $\propto t$ and $\propto J$ renormalized by what appears to be a similar scale $\alpha \approx 0.33$? Employing the string-picture described in the paragraph above, the main source of renormalization of the bare

dispersions comes from a competition of kinetic and potential energy of the string in the two-hole bound state. This depends only on the internal structure of the pair, which we believe to be quantitatively extremely similar in the heavy and light types of paired states (with dispersions $\propto J$ and $\propto t$, respectively), see Fig. 7 in [Grusdt et al., SciPost Phys., SciPost, **2023**, 14, 090] for a quantitative analysis. Hence, we expect that the overall dispersion of each pair, dominated by either t or J , should be rescaled by a similar amount α (essentially a Franck-Condon factor describing the overlap of fluctuating string states formed around different sites), in excellent agreement with our numerical findings.

To address this comment by the referee, we have clarified our statement in the second-to last paragraph of the introduction, where we emphasize that a main and surprising finding of our work is the existence of a long-lived two-hole bound state with a dispersion $\propto t$, which does not directly decay into individual single hole states.

Referee:

- Is the hole doping rate $p=2/(4*40)=1.25\%$? Does the system retain antiferromagnetic order? Do the authors consider a coexistence state of pairing and antiferromagnetism? Answers should be given in the main text.

Our reply:

In our work we consider two-hole spectra in an *initially undoped* antiferromagnet at $\delta = 0$ hole doping. We believe it is not justified (or useful) to consider the dynamically evolving system that we study, with the two holes added, as being at finite-doping in any sense, for several reasons:

- (i) We chose the long length $L_x = 40$ of our numerically studied system in such a way that the wavefronts of the hole-pair created in the center around $x = 20$ at time $t_0 = 0$ does not reach the edges within the finite evolution time $t_1 = T$ that we are able to simulate. While this Fourier-limits our spectral resolution, it means finite-size effects $\simeq 1/L_x$ along the extended direction can be essentially neglected, and at the given spectral resolution we would expect identical results had we worked in systems with $L_x \gg 40$. Therefore our results are best described as Fourier-limited two-hole spectra on top of an **undoped** antiferromagnet.
- (ii) In general, we study a dynamical response, and within the underlying evolution time T the single pair of holes cannot globally equilibrate. For example, the density distribution of the holes does not reach any steady-state value, which means that no meaningful definition of the doping-level can be assigned.
- (iii) Of course the *eigenstates* of the two-hole system we consider may contain interesting states in which the presence of the two holes may possibly destroy the antiferromagnetism, and in this case an interpretation based on some non-zero average doping value δ , probably around $\delta = 2/(4 \times 40) = 1.25\%$ as suggested by the referee, may be meaningful. However, the observable we extract is the two-hole spectral function: As becomes clear from its Lehmann-representation, see Eq. (6) in our manuscript, only two-hole eigenstates $|\Psi_n\rangle$ contribute to the spectrum which have appreciable spectral weight, defined by the overlap $|\langle \Psi_n | \hat{\Delta} | \Psi_0 \rangle|^2$, where $\hat{\Delta}$ creates a pair of two tightly-bound holes on top of the original undoped antiferromagnet $|\Psi_0\rangle$. Therefore, eigenstates with two holes but no long-range AFM order (e.g. in a setting where the two holes in the finite-size setting bind to a domain-wall of the antiferromagnet) will contribute negligibly small spectral weight owing to the extended size $L_x = 40$ of our numerically studied system.

We thank the referee for raising this important point and agree that an explicit discussion in the main text would be beneficial. Hence we have included a discussion below Eq. (1) in our manuscript.

Referee:

- How does one understand the absolute energy in, for example, Fig.1? I first assumed that this energy should be related with the pairing gap energy, but this should not be the case since the energy scale is too high in Fig.1. A physical understanding of the absolute energy should have been given.

Our reply:

The energies ω in the spectra are measured relative to the energy of the undoped parent antiferromagnet. This is apparent from the definition of the two-hole Green's function in Eq. (5) of our manuscript and becomes explicit in its Lehmann representation, see Eq. (6) in our manuscript: There one sees clearly that delta-function peaks appear at frequencies $\omega = E_n - E_0^0$ where E_n is the eigenenergy of a given two-hole eigenstate and E_0^0 is the ground state energy of the initial undoped antiferromagnet.

We agree with the referee, however, that this can be mentioned even more explicitly. Hence we have added a sentence to the caption of Figure 1 in the revised manuscript: "*Energies are measured relative to the undoped parent antiferromagnet.*"

A more quantitative analysis suggests that the overall energy scale per hole should be $E_0^{1h} - E_0^0 \simeq -2\sqrt{3}t + AJ^{2/3}t^{1/3} + \mathcal{O}(J)$, with some numerical coefficient A ; the leading term $-2\sqrt{3}t = -10.3\dots J$ explains the overall scale, consistent with Fig. 1. See e.g. Ref. [Grusdt et al., SciPost Phys., SciPost, **2023**, 14, 090] for discussions / derivations of these estimates.

The referee is also asking for a physical understanding of the absolute energy scale: This is simply the amount of energy it takes to create a pair of holes in the original antiferromagnet. It is negative, because the overall energy is dominated by a kinetic energy contribution $\simeq -t$ per hole. The referee correctly notes that the binding energy is much smaller: to extract it, one has to compare twice the energy cost $E_0^{1h} - E_0^0$ for adding a single hole to once the energy cost of adding two holes simultaneously $E_0^{2h} - E_0^0$. The referee correctly concludes that the resulting binding energy $E_{\text{bdg}} = E_0^{2h} - E_0^0 - 2 \times (E_0^{1h} - E_0^0)$ is much smaller (well below J) than the overall scale on which the two-hole spectra in Figs. 1, 2 of our manuscript show pronounced coherent quasiparticle features.

We note that the scale of the binding energy can be explicitly read off in Fig. 2 a) of our manuscript: there we plot twice the energy of a single hole as a green dashed line, as explicitly indicated in the figure caption. To emphasize this point, we have explicitly added a clarifying statement in the caption of Fig. 2: "*indicating a small pairing gap on the order of Jz* ".

Referee:

- Rather the width of the dispersion scaled by J seems to be related with the pairing gap amplitude. On the other hand, the width of the dispersion scaled by t may not have a physical meaning of the pairing gap. A physical understanding of the dispersion width should have been given.

Our reply:

We respectfully disagree with the referee that the width of the dispersion is related with the pairing gap amplitude (the latter being essentially the *binding energy*). As explained in our response to the previous point, the binding energy must be obtained by comparing the lowest-energy peak in the one-hole spectral function (or, simpler and numerically more reliable, ground state energy in the one-hole sector) to the lowest-energy peak in the two-hole spectrum that we show. The bandwidth of a given dispersive feature in the spectrum, in contrast, does not allow for any direct conclusions about the binding energy.

To address the referee's request for a physical understanding of the pair dispersion, we would like to mention three interesting points:

- (i) The total dispersion of a pair is not often discussed in the context of pairing, partly because in conventional BCS superconductors the Cooper pairs simply condense in their lowest-energy momentum state and the pair-dispersion rarely plays any specific role. Moreover, the pair dispersion in a conventional BCS superconductor is not all that different from the dispersion of the individual fermions. This is in sharp contrast, however, to the situation in strongly correlated settings like the one we study in our manuscript: As we demonstrate, the pair dispersion shows two distinctly different branches — one heavy pair with dispersion $\propto J$, and one light pair with dispersion $\propto t$ — both of which differ significantly from the dispersion of individual holes alone. This points to the rich and non-trivial underlying structure of the pairs, which we aim to understand in order to get a better microscopic understanding of the underlying pairing mechanism(s) at play.
- (ii) A main result of our manuscript is the good qualitative agreement we observe between our unbiased numerical simulations of the two-hole spectrum and predictions by an effective theory of pairs of holes tightly bound by a string, see Figs. 2 and 3 and supplementary material. This means the effective string theory provides very useful physical understanding of the observed pair dispersions. We agree with the referee that a more easily accessible discussion of the effective string theory would be useful, and as also proposed by Reviewer #1, we have decided to add a brief but self-contained summary in the method's section of the revised manuscript. The main reason why we originally hesitated to discuss the somewhat phenomenological string theory in any detail was our desire to present our numerical results in a completely unbiased way.
- (iii) There exists a very simple and intuitive explanation of the general features we observe: The highly dispersive two-hole band can be understood by assuming a tightly bound pair of holes connected by a string of displaced spins in the host antiferromagnet; one hole can follow the other by re-tracing the string, which allows the pair to move through the medium almost freely, with an overall dispersion $\sim t$. The flat bands can be attributed to similar tightly bound pair of holes connected by a string, but with non-trivial rotational symmetry; the latter leads to constructive interference when the pair moves, which completely localizes the center-of-mass motion of the pair; see our section in the supplementary material for more details.

That said, we now understand that these points are difficult for our readers to extract from the main text in our manuscript, and thank the reviewer for drawing our attention to a lack of a simple and intuitive explanation of the basic physics. To address this point, we have decided to add a slightly extended paragraph describing our interpretation in terms of the geometric string theory at the end of Sec. "The t-XXZ model": *"The geometric string theory correctly predicts the highly dispersive peak, as well as the existence of completely flat bands. Within this effective theory, the highly dispersive peak can be attributed to configurations where one hole re-traces the string created by the other hole, which allows the pair to move freely through the host antiferromagnet*

with an overall dispersion $\sim t$. The completely flat bands we find have also been predicted by an earlier theoretical study using a similar effective model [Shraiman1988]. We attribute them to destructive interference of hole pairs with non-trivial rotational symmetry. A self-contained summary of the effective string theory is provided in the methods section."

Referee:

- In the lightly doped t-J model, the Fermi surface may appear as a hole pocket around $(\pi/2, \pi/2)$. Therefore one would expect that pairing should occur there, which is, however, not the case in Fig.1. Why?

Our reply:

We thank the referee for raising this important question. The essence of the answer is as follows: The tightly bound pairs we observe in the two-hole spectral function *cannot* be understood by a scenario where magnetic polarons — strongly renormalized individual holes, with dispersion minima around $(\pi/2, \pi/2)$ in the t-J model as alluded to by the referee — form first, which then pair up in a second step. Instead, the holes **form a new type of paired state** which is not directly related to the state formed by individual holes. This can be seen most readily by the emergence of a wide dispersion $\sim t$, and is also directly related to the order in which spin-exchange and tunneling terms in the Hamiltonian are renormalized, see our reply above to an earlier point raised by Reviewer #2.

Let us add a more concrete discussion to support our argument:

- (i) In our manuscript, we start by analyzing the t-XXZ model with strong Ising anisotropy. In this case, the individual holes do *not* form pockets around $(\pi/2, \pi/2)$; instead, the lowest-energy state of a single hole in the Ising background carries momentum (π, π) and $(0, 0)$, see e.g. [Grusdt et al., PRX 8, 011046 (2018)]. Then we move on to study the SU(2) invariant t-J model, which indeed has lowest-energy single-hole states around the nodal points $(\pi/2, \pi/2)$. Nevertheless, the pair-dispersions we observe in the two-hole spectra remain largely unchanged: As we demonstrate in Fig. 3, the width of the highly dispersive feature does not change in any meaningful way; as we demonstrate in Fig. 4, the flat bands from the Ising limit develop a weak dispersion on the order of $\sim J_{\perp}^2/J_z$, always staying well below the distinctly wider width $\sim t$ of the light hole pair. This numerically establishes that the structure of the pairs can remain largely independent of the properties of the individual single-hole ground state.
- (ii) An intuitive explanation of the above-mentioned behavior can be obtained from the parton picture of doped holes, which was originally proposed in [Béran et al., Nuclear Physics B, 473, 707 (1996)] and further refined and confirmed numerically later on, see e.g. [Bohrdt et al., Phys. Rev. B 102, 035139 (2020)]. In essence, it describes the individual holes as being bound states of two (confined) partons: a light spin-less chargon (or holon) and a heavy charge-neutral spinon. The center-of-mass dispersion of this bound state, i.e. of the single magnetic polaron, is dominated by the dispersion of the heavier of the two constituents, i.e. for $t > J$ which we consider, by the spinon; hence the magnetic polaron dispersion $\sim J$ emerges. Moreover one can understand why magnetic polarons form hole-pockets around $(\pi/2, \pi/2)$ in the SU(2)-invariant t-J model, and around $(0, 0)$ in the Ising case. Within the parton picture, the robust paired states we numerically establish in our present manuscript should be understood as *just chargon-chargon bound states*, which do not involve spinons at all. This explains why a robust broad dispersion $\sim t$ emerges, which remains essentially unchanged as the properties of the spin-sector is changed (e.g. by going from t-Jz to the t-J model).

(iii) That said, one can imagine forming less tightly bound states of magnetic polarons. Their parton description involves an overall bound state of *four* partons: two spinons and two chargons. In this case, each hole remains bound to a spinon and the properties of the overall pair are expected to correlate strongly with those of the one-hole ground state. In particular, the center-of-mass dispersion of this type of hole pair is expected to change significantly when going from the t-J_z to the t-J model. As detailed in our reply to the last point raised by Reviewer 1, we believe that such kinds of tetra-parton bound states, i.e. Cooper pairs of magnetic polarons, may play a direct role for understanding pairing in high-T_c cuprates, and the underlying attractive interaction between these magnetic polarons may be explained as being mediated by the much more tightly bound chargin-chargin pairs that we discovered in this manuscript. Delineating one type of pairing from another in a most realistic model will be an important subject of future research.

We believe that the discussion provided here for Reviewer 2 partly goes beyond the scope of our present manuscript, but find the comment raised by the referee important nevertheless. In order to address it in our revised manuscript, we have added a sentence at the end of the "Summary & Outlook" section, stating: *"Furthermore, our work raises the interesting question how the two types of tightly bound hole pairs discovered here relate to the Cooper pairs constituting high-temperature superconductors in copper oxides."*

Referee:

- Just below Fig.2 the authors state "Remarkably, the two-hole spectral function, Fig.2, exhibits a highly dispersive peak with a mass proportional to 1/t.". I am not sure which panel in Fig.2 the authors have in mind. $k_y=\pi$?

Our reply:

As the referee suspects, this highly dispersive feature is best seen in the bottom panel of Fig. 2, i.e. at $k_y=\pi$. In the bottom panel, the black line corresponds to a cosine-fit to the dispersive feature, the bandwidth of which is established to scale $\sim t$ in Fig. 3 by comparing different ratios of t/J_z . To clarify this in the manuscript, we have added in the main text: *"..., best identified at $k_y=\pi$ (bottom panel)"*.

However, the highly dispersive feature is also visible at $k_y=\pi/2$ (in the middle panel of the old Fig. 2 — in the revised Fig. 2 we removed $k_y=\pi/2$ since it contains no qualitatively new information; the data at $k_y=\pi/2$ is still presented in the supplementary material): at low energies (k_x around π) we observe a well-defined peak in the two-hole spectrum: The latter closely resembles our prediction by the geometric string theory, in which the dispersion scales as $\sim t$. For smaller k_x around 0, the feature quickly loses spectral weight and appears to merge with a higher-energy continuum of additional states.

Referee:

- What is ω_{\max} ? It can be positive in Fig.3 and negative in Fig.4.

Our reply:

We extracted the peak position $\omega_{\max}(\mathbf{k})$ from spectra as shown in Fig. 2. Since our primary goal in Fig. 3 and 4. is to compare bandwidth and the shape of the dispersions, overall shifts $\varepsilon(t/J_{(z)})$ independent of momentum \mathbf{k} were added to the curves:

$$\omega_{\max}(\mathbf{k}, t/J_{(z)}) \rightarrow \omega_{\max}(\mathbf{k}, t/J_{(z)}) + \varepsilon(t/J_{(z)}) = \Delta\omega_{\max}(\mathbf{k}, t/J_{(z)})$$

In Figs. 3 and 4 we directly plot $\Delta\omega_{\max}(\mathbf{k}, t/J_{(z)})$ for different values of t/J and t/J_z . Since we originally chose an arbitrary overall reference energy, the absolute values on the y-axis of Figs. 3 and 4 are not meaningful — in particular the fact that one is positive while the other is negative has no significance.

We thank the reviewer for drawing our attention to this non-ideal presentation. In the revised version of our manuscript we have changed the labels on the y-axes in Figs. 3 and 4 to $\Delta\omega_{\max}/t$ to clarify that we consider differences. Moreover, we have chosen the overall offsets such that the dispersion collapses at $k_x=0$ which we choose as an (arbitrary) reference point; i.e. the overall scale on the y-axis meaningfully indicates the overall scale of the bandwidth in units of the tunneling t .

Referee:

- I do not figure out a reason why the lowest lying peak for $k_y=0, \pi/2$ becomes completely flat whereas that for $k_y=\pi$ does not.

Our reply:

We believe this has to do with the fact that the flat bound state at $k_y=\pi$ has vanishing spectral weight, and thus cannot be seen in the two-hole spectrum.

This intuition is supported by the effective string theory results presented in the supplementary material: In Figs. SM1 and SM3 we show s-wave and d-wave spectra expected from the geometric string theory — these figures are re-printed here for convenience:

s-wave spectrum in Supp.Mat. Fig. 1: Left: effective string theory, right: DMRG simulations on 4-leg cylinder.

FIG. 1. Comparison of s-wave spectra to geometric string theory prediction, [1], left, to numerically calculated spectra (right), for the $t - J_z$ model with $t/J_z = 3$, $\chi = 600$, time evolution up to $T_{max}/J_z = 10$, on a 40×4 cylinder. Top, middle, bottom plots correspond to momentum $k_y = 0, \pi/2, \pi$, respectively.

d-wave spectrum in Supp.Mat. Fig. 3: Left: effective string theory, right: DMRG simulations on 4-leg cylinder.

FIG. 3. Comparison of *d*-wave spectra to geometric string theory prediction, [1], left, to numerically calculated spectra (right), for the $t - J_z$ model with $t/J_z = 3$, $\chi = 600$, time evolution up to $T_{max}/J_z = 10$, on a 40×4 cylinder. Top, middle, bottom plots correspond to momentum $k_y = 0, \pi/2, \pi$, respectively.

To understand how this relates to Fig. 1 in the main text of our manuscript, we note the following:

- (i) From the geometric string theory, we don't expect a flat band in the s-wave channel, see Fig.1 in the SM. However, we do expect a flat band with non-vanishing spectral weight in the d-wave channel, see Fig. 3 in the SM.
- (ii) Notably, at $k_y = \pi$, neither in d-wave nor in s-wave does the geometric string theory predict any spectral weight in the lowest-lying flat band. This is consistent with our numerical observation that there is no low-lying flat band at $k_y = \pi$, as pointed out by Reviewer 2.

This observation, among others, demonstrates the predictive power of the geometric string theory. While this comparison should not be taken as a proof of any sort, we find that it provides a compelling explanation of a curious numerical observation.

Referee:

- In Fig.3 I do not see "yellow crosses".

Our reply:

We apologize if any graphics error has occurred in the figure. We re-print here the revised Fig. 3, highlighting the positions of the yellow crosses in the right panel:

In summary, we thank the reviewer again for taking time to carefully consider our manuscript, for raising several important and interesting points and for not judging our work prematurely. We hope we could address all questions raised by the reviewer satisfactorily and hope they can follow the recommendation of Reviewer 1 and support publication of our revised manuscript in Nature Communications.

List of changes:

- We have fixed a bug that we have identified after receiving the referee reports in the evaluation of the rotational spectra (the raw data was correct and has not been changed, but to calculate the rotational Green's function different contributions need to be added up with correct phases, and this is where a small numerical mistake was previously made). As a result we have:
 - updated Fig. 1 (b), fixing the bug in the numerical evaluation. Now we show only results for $k_y=0$ (for d-wave) and $k_y=\pi$ (for s-wave) for clarity.
 - We have updated Fig. 2, fixing the bug in the numerical evaluation. Now we show only results for $k_y=0$ (for d-wave) and $k_y=\pi$ (for s-wave) for clarity.
 - We have updated Fig. 5, fixing the bug in the numerical evaluation. Now significantly less hybridization is visible between s-wave and d-wave excitations. This allows us to distinguish the two types of excitations much more clearly in our numerics; we have modified the text describing Fig. 5 accordingly.
 - We have corrected all figures affected by the bug in the supplementary material.
- In the section "The t-XXZ model" we added a reference to a recent theoretical proposal how to realize this model using Rydberg atoms or ultracold molecules in tweezer arrays.
- We updated the reference [Grusdt, Demler & Bohrdt, arXiv:2210.02321] which has now been published as: [Grusdt et al., SciPost Phys., SciPost 14, 090 (2023)].
- In the second paragraph of the outlook & summary, we clarified that we are referring to ARPES experiments in quantum simulators, which have been carried out using ultracold atoms.
- Following the suggestion of Referee 1, we have included a methods section which provides a self-contained overview of the geometric string theory — an effective semi-analytical approach to which we compare our large-scale numerical results.
- In the summary & outlook section, following the advise by Referee 1, we have added a paragraph: *"Our numerical studies of spectra are currently limited to four-leg systems, since we cannot reach sufficiently long times to achieve the desired spectral resolution when working with larger bond-dimensions required for six-leg systems. Finite-size effects can be expected to play a role, in particular on a quantitative level, but the good qualitative agreement of our results with predictions by the genuinely two-dimensional geometric string approach, along with the absence of significant hybridization of d and s wave excitations, support the view that four-leg cylinders are sufficient to capture key qualitative properties of hole pairs."*
- Addressing a point raised by Referee 2, we have modified the second-to last paragraph of the introduction section to clarify that the most remarkable observation we make is related to the existence of a coherent peak in the two-hole spectrum extending over a range of energies set by the large tunneling t instead of being renormalized down to J scales as usually happens in the single-hole doped case. We have also added a reference to the seminal paper [Kane et al., Phys. Rev. B 39, 6880-6897 (1989)] treating the single-hole case.
- Addressing another point raised by Referee 2, we have clarified throughout the manuscript that we work in the zero-doping limit, and our system cannot be meaningfully considered as being at non-zero doping. This includes an explicit discussion that we have added below Eq. (1).
- Following the advise by Referee 2, we have added an explicit statement in the caption of Fig. 1 that: *"Energies are measured relative to the undoped parent antiferromagnet."*
- To clarify another question raised by Referee 2, we have added a statement in the caption of Fig. 2: *"indicating a small pairing gap on the order of Jz ".*
- To address Referee 2's request for a simple explanation of the observed dispersion relations, we have decided to add a slightly extended paragraph describing our interpretation in terms of the geometric string theory at the end of Sec. "The t-XXZ model": *"The geometric string theory*

correctly predicts the highly dispersive peak, as well as the existence of completely flat bands. Within this effective theory, the highly dispersive peak can be attributed to configurations where one hole re-traces the string created by the other hole, which allows the pair to move freely through the host antiferromagnet with an overall dispersion $\sim t$. The completely flat bands we find have also been predicted by an earlier theoretical study using a similar effective model [Shraiman1988]. We attribute them to destructive interference of hole pairs with non-trivial rotational symmetry. A self-contained summary of the effective string theory is provided in the methods section."

- To address a point raised by Referee 2, we have added a sentence at the end of the "Summary & Outlook" section: "*Furthermore, our work raises the interesting question how the two types of tightly bound hole pairs discovered here relate to the Cooper pairs constituting high-temperature superconductors in copper oxides.*"
- Following the advise by Referee 2, we have adapted the y-axes labels in Figs. 3 and 4 to clarify that we plot energy differences here rather than overall energies.
- We have fixed a typo in Eq. (2) — an extra factor of $\pi/2$ was included in the last version.
- We have included a reference to Fig. 1 a) below Eq. (2) to better illustrate what is measured.
- We fixed a typo in the legend of Fig. 2: the geometric string lines have been shifted by $-0.35 Jz$ (not $-0.35 J$).

REVIEWERS' COMMENTS

Reviewer #1 (Remarks to the Author):

I have gone through the detailed response by the authors to both referees and the list of changes to the manuscript. I think the paper can now be accepted for publication.

Reviewer #2 (Remarks to the Author):

I have reviewed the revised manuscript as well as the authors' response. Thanks to the authors' explanation, I understand which parts I misunderstood in the first round and now get much sharper understanding of the authors' work. In this sense, the authors' revisions are successful. I think that the present work is sound scientifically and can be published in some journal.

If I am asked whether this manuscript should be published in Nat. Commun., I would be negative and would suggest a more sophisticated journal. This is because the present work does not touch directly crucial insight into the pairing mechanism in real materials, but less-sharp insight into that (although theoretical calculations themselves are clear now as I mentioned above); a potentially important role of highly mobile hole pairs, which is a key finding in this work, is not so clear in the context of the pairing mechanism; with a sharper understanding of the authors' work eventually, I think it interesting, but I consider that this work has a merit mainly to experts working in the same field as the authors and thus the scope of the present work is rather limited. The authors surely have the opposite opinion to mine, but I am not convinced in the second round.